# Clinical Strains of *Mycobacterium tuberculosis* Representing Different Genotype Families Exhibit Distinct Propensities to Adopt the Differentially Culturable State

**DOI:** 10.3390/pathogens13040318

**Published:** 2024-04-12

**Authors:** Bhavna Gowan Gordhan, Kiyasha Padarath, Astika Sewcharran, Amanda McIvor, Michael S. VanNieuwenhze, Ziyaad Waja, Neil Martinson, Bavesh Davandra Kana

**Affiliations:** 1Department of Science and Innovation and the National Research Foundation Centre of Excellence for Biomedical TB Research, School of Pathology, Faculty of Health Sciences, University of the Witwatersrand, Johannesburg 2017, South Africa; bhavna.gordhan@nhls.ac.za (B.G.G.); kiyasha.padarath@wits.ac.za (K.P.); astika.sewcharran@nhls.ac.za (A.S.); amanda.axcell@gmail.com (A.M.); 2National Health Laboratory Service, Johannesburg 2000, South Africa; 3Department of Chemistry, Indiana University Bloomington, Bloomington, IN 47405-7000, USA; mvannieu@iu.edu; 4Perinatal HIV Research Unit (PHRU), University of the Witwatersrand, Johannesburg 2017, South Africa; wajaz@phru.co.za (Z.W.); martinson@phru.co.za (N.M.); 5Center for TB Research, Johns Hopkins University, Baltimore, MD 21218, USA

**Keywords:** differentially culturable tubercle bacteria, tuberculosis, *Mycobacterium tuberculosis*, Beijing strains, LAM strains

## Abstract

Growing evidence points to the presence of differentially culturable tubercle bacteria (DCTB) in clinical specimens from individuals with active tuberculosis (TB) disease. These bacteria are unable to grow on solid media but can resuscitate in liquid media. Given the epidemiological success of certain clinical genotype families of *Mycobacterium tuberculosis*, we hypothesize that different strains may have distinct mechanisms of adaptation and tolerance. We used an in vitro carbon starvation model to determine the propensity of strains from lineages 2 and 4 that included the Beijing and LAM families respectively, to generate DCTB. Beijing strains were associated with a greater propensity to produce DCTB compared to LAM strains. Furthermore, LAM strains required culture filtrate (CF) for resuscitation whilst starved Beijing strains were not dependent on CF. Moreover, Beijing strains showed improved resuscitation with cognate CF, suggesting the presence of unique growth stimulatory molecules in this family. Analysis of starved Beijing and LAM strains showed longer cells, which with resuscitation were restored to a shorter length. Cell wall staining with fluorescent D-amino acids identified strain-specific incorporation patterns, indicating that cell surface remodeling during resuscitation was distinct between clinical strains. Collectively, our data demonstrate that *M. tuberculosis* clinical strains from different genotype lineages have differential propensities to generate DCTB, which may have implications for TB treatment success.

## 1. Introduction

Several studies have demonstrated the presence of differentially culturable/detectable tubercle bacteria (DCTB/DDTB) in sputum, which display the capacity for growth only in liquid and not on solid media [1,2,3,4,5]. A hallmark of DCTB is their ability to grow in the presence of culture filtrate (CF), which contains growth stimulatory factors including resuscitation-promoting factors (Rpfs) that enhance the growth of non-replicating bacteria [1,2,3,4,5,6,7,8]. Current tuberculosis (TB) diagnostic assays likely miss DCTB, resulting in diagnostic inaccuracies. This problem can be exacerbated in certain vulnerable populations, such as people living with HIV and children, who are often unable to produce sputum or have low bacterial loads in their clinical specimens, thus presenting as smear negative during diagnosis. This underscores the importance for improved diagnostics to detect low bacterial loads, particularly if the prevailing bacteria are DCTB.

During TB infection, mycobacteria typically adapt to hostile conditions within the lung granuloma, such as carbon and oxygen starvation, by entering a non-replicating state characterized by metabolic remodeling and increased antibiotic tolerance [9,10,11,12,13]. Global control of TB therefore requires interventions that effectively address detection of all adapted populations of *M. tuberculosis* during TB pathogenesis. Deeper analysis of the phenotypes of such cells would greatly enhance the development of simpler methods of enumerating them and developing drugs that could aid their eradication.

In addition to detecting conventionally culturable bacteria, addressing the evolution of drug-resistant strains requires the development of appropriate clinical endpoints that enable the quantification of DCTB/DDTB, which display drug tolerance [5,8,14]. This will allow for the assessment of newer regimens to eradicate persister bacteria, thus effectively shortening treatment duration with rapid elimination of all populations of mycobacteria. In a recent study, sputum samples from individuals with drug-resistant TB (DR-TB) on appropriate DR-TB treatment showed the presence of DCTB up to 16 weeks post treatment, suggesting that DCTB may play a role in the interpretation of early bacterial assay (EBA) trials of investigational TB drugs for hastening sputum conversion after treatment [4]. Similar results were obtained with drug-susceptible tuberculosis using standard therapy [5]. Therefore, methods for generating DCTB in vitro could aid in understanding metabolic processes that enable tubercle bacilli to persist.

Several in vitro models mimicking physiological stresses encountered within the granuloma have been developed for generating non-replicating *M. tuberculosis* with phenotypic tolerance to antimicrobial drugs, whilst retaining the ability to resume growth upon removal of the stress [15,16,17,18,19,20,21,22,23]. Betts and co-workers [15] developed an in vitro nutrient starvation model for *M. tuberculosis*, based on a model described 70 years ago by Loebel et al. [15,17]. In this model of abrupt carbon starvation, bacteria undergo growth arrest and decreased respiration rate, and become increasingly drug tolerant while maintaining viability, thus simulating *M. tuberculosis* in a persistent state. More recently, an in vitro model utilizing nutrient starvation followed by exposure to rifampin (RIF), a first-line TB drug, produced DCTB and demonstrated the utility of this system for identifying compounds that can kill this bacterial population [23]. Whilst these models have been useful to study metabolic changes during non-replicating persistence of bacteria, how these effects prevail in different clinical strains of *M. tuberculosis* requires further investigation.

In South Africa, the TB epidemic is dominated by four genotype families of *M. tuberculosis*, two of which, the Beijing and F15/LAM4/KZN genotype families, are the subject of intensive study to understand the basis for their high penetration in humans [24]. The F15/LAM4/KZN family has been associated with multiple drug resistance and was responsible for the first described case of extensively drug-resistant TB in South Africa [25]. Phylogenetic analysis showed that the Beijing genotype family can be grouped into at least seven different sub-lineages with variable virulence that correlate with their ability to spread and cause disease [26]. The Beijing genotype family, compared to other family types, tends to be more transmissible and have an increased inherent propensity to evolve drug resistance [27]. Furthermore, the Beijing family has an increased expression of dormancy regulon genes under hypoxic stress conditions, suggesting that these strains are well adapted during TB pathogenesis [28]. However, little is known about the capability of different genotype families of *M. tuberculosis* to generate DCTB.

In this study, we used an in vitro carbon depletion model to assess the propensity of strains from the Beijing and LAM families to generate DCTB and their subsequent ability to resuscitate under nutrient-rich conditions. In addition to using limiting dilution assays with CF, resuscitation of DCTB was also assessed by fluorescent microscopy with the TADA (TAMRA-3-amino-D-alanine) probe, which incorporates into the cell wall of actively replicating cells [29]. We further investigated the dependency on CF, in addition to examining modalities of cell wall expansion during resuscitation.

## 2. Materials and Methods

All methods were conducted in accordance with the relevant regulations related to handling human specimens and the guidelines for growth of *M. tuberculosis*. All procedures were conducted in a Biosafety Level III laboratory, registered with the South African Department of Agriculture Forestry and Fisheries (registration number: 39.2/NHLS-20/010) and approved by the Institutional Biosafety Committee of the University of the Witwatersrand (approval number: 20200502Lab).

### 2.1. Assessment of DCTB in Clinical Cohorts

The DCTB data generated from two previous clinical studies carried out in our laboratory, named the BMG study and the MGIT-Plus study, were retrospectively analyzed in this study [5,30]. Ethics approval for both studies was obtained from the Human Research ethics committee of the University of Witwatersrand (protocol numbers: M140265 (BMG) and M110833 (MGIT-Plus)). The combined number of participants analyzed from the two studies was 160. DNA extracted from bacteria isolated and stored from the sputum of these participants, during the assessment of differentially culturable tubercle bacilli (DCTB), was subjected to spoligotyping as described below, to classify the mycobacteria into the different *M. tuberculosis* genotype families. Genotype families with 10 or less strains were excluded (n = 36). Different genotype families from the remaining samples (n = 124) were separated into two categories: (i) participants that produced DCTB under any one of the MPN assay conditions, i.e., media plus the addition of growth factors in the form of culture filtrate (CF) from an axenic culture of *M. tuberculosis,* H37Rv (CF-dependent) or media only (CF-independent); and (ii) those that did not produce DCTB. In the latter case, the number of colony-forming units (CFUs) representative of culturable bacteria was higher than the growth measured in the MPN assay. The ratio of DCTB to no DCTB was calculated for samples in each genotype family. Next, samples within each genotype family were analyzed for the number of specimens that produced DCTB (n = 66), under both CF-independent and CF-dependent conditions, and those that did not (n = 43). This is outlined in Figure 1A.

### 2.2. Spoligotyping of Mycobacterium tuberculosis Strains

The kit and reagents for spoligotyping the strains were supplied by MapMyGenome (Hyderabad, India) and the protocol instructions provided by Ocimum Biosolutions (Hyderabad, India) were followed. The unique 43 spacer DNA sequences, depicting a different *M. tuberculosis* family within the direct repeat (DR) region, specifically present in the *M. tuberculosis* complex, were amplified using primers designed for this region [31]. A master mix comprising 500 µL of the Kapa master mix (Roche holding AG, Basel, Switzerland), 260 µL of Milli Q water (Separations Scientific, Johannesburg, South Africa) and 120 µL each of DR A and DR B primers was prepared. Twenty-three microliters of this solution was added to each PCR tube, together with 5 µL of genomic DNA. Cycling conditions (30×) were as follows: initial denaturation at 95 °C for 3 min, denaturation at 94 °C for 1 min, annealing of primers at 55 °C for 1 min, extension at 72 °C for 30 s, with a final extension at 72 °C for 10 min. The membrane was hybridized in prewarmed 2× SSPE/0.1% SDS solution at 60 °C for 1 h prior to loading the PCR samples. A 150 µL aliquot of the 2× SSPE/0.1% SDS solution was added to each PCR product and the DNA denatured at 99 °C for 10 min. The tubes were placed on ice to prevent annealing of the DNA strands. The denatured PCR products were loaded onto the pre-hybridized membrane assembled in an array blotter, together with the positive controls (DNA from H37Rv and BCG, supplied by MapMyGenome) and negative control (no template control and 2× SSPE/0.1% SDS solution). The array blotter was incubated for an hour at 60 °C before washing the membrane with 2× SSPE/0.5% SDS for 5 min at 60 °C to remove any unbound probe. The washed membrane was then incubated in 40 mL of 2× SSPE/0.5% SDS containing 10 µL of Streptavidin-HRP conjugate (Thermo Fisher Scientific, Waltham, MA, USA) which binds to the biotin label on the PCR products. The membrane was incubated at 42 °C for 1 h and then washed twice for 5 min with 2× SSPE/0.5% SDS at 42 °C and twice for 5 min with 2× SSPE at room temperature. The signals were detected with 10 mL each of enhanced chemiluminescence (ECL) detection solutions 1 and 2 and the emitted light from the reaction was developed by autoradiography of the membrane. The presence or absence of the 43 unique spacers in the DR region was used to determine the strain type of each *M. tuberculosis* sample using the online software tool SpotClust https://tbinsight.cs.rpi.edu/run_tb_lineage.html or https://tbinsight.cs.rpi.edu/complex_families.html, software developed by NIH and Rensselaer Polytechnic Institute. [31]. First software accessed on 28 November 2014 and the second link was accessed on 14 March 2024.

### 2.3. Mycobacterial Culture Conditions

*M. tuberculosis* H37Rv and the clinical Beijing and LAM genotype families used in this study were grown in Middlebrook 7H9 (Becton-Dickenson, Franklin Lakes, NJ, USA) media supplemented with 0.2% glycerol (Sigma, Springfield, MO, USA), 10% Middlebrook oleic acid-albumin-dextrose-catalase (OADC, Becton-Dickenson) and 0.05% Tween 80 (Sigma, Missouri, USA) to mid-log phase (OD_600nm_ 0.5–0.8) at 37 °C without shaking. Five individual strains from the Beijing family and five from the LAM family were thawed from −80 °C freezer stocks, streaked on BBL™ Middlebrook 7H11 (Sigma, Springfield, MO, USA) media and incubated for 3 weeks at 37 °C. Single colonies were selected and sub-cultured for 7 days at 37 °C in 5 mL of Middlebrook 7H9 media supplemented with OADC and Polymyxin B, Amphotericin B, Nalidixic acid, Trimethoprim and Azlocillin (PANTA, Becton-Dickenson, Franklin Lakes, NJ, USA) reconstituted in OADC, to ensure the strains were free of contaminants. Once the cultures reached mid-log phase (OD_600nm_ 0.5–0.8), the cells were used as a preculture to regrow the strains, again in Middlebrook 7H9 media supplemented with OADC and PANTA, to ensure any remaining contamination was killed. The mid-log liquid culture was then spread on BBL™ Middlebrook 7H11 plates supplemented with OADC to confirm that the Beijing and LAM strains were free of contamination.

### 2.4. Growth Kinetics of Mycobacterial Strains

Single colonies of H37Rv, the Beijing genotype family and the LAM genotype family were inoculated into 5 mL of supplemented 7H9 media and grown to mid-log phase (OD_600nm_ = 0.5–0.8). A 200 μL aliquot of the pre-culture was then inoculated into 20 mL of fresh 7H9 to an approximate OD_600nm_ = 0.05 and incubated at 37 °C without shaking. Growth was monitored by taking OD_600nm_ measurements every 24 h over a 15-day period.

### 2.5. In Vitro Carbon Starvation Model for the Generation of Viable but Non-Culturable M. tuberculosis

An in vitro *M. tuberculosis* carbon starvation model was developed based on that described by Betts and colleagues [15]. A subsequent publication using the combination of carbon starvation and rifampicin treatment was shown to generate DCTB; hence, we used this model to generate DCTB [23]. Briefly, the *M. tuberculosis* laboratory strain (H37Rv) and the 5 strains each from the Beijing and LAM genotype families were grown in triplicate in 20 mL of 7H9 Middlebrook media to an optical density (OD_600nm_) of 0.5–0.8, corresponding to 10^5^–10^9^ CFU/mL. The cultures were centrifuged at 4000× *g* for 10 min and washed twice with 20 mL of phosphate buffer saline (Sigma, Springfield, MO, USA) containing 0.05% tyloxapol (Sigma, Springfield, MO, USA) (PBS-Tx). After the second wash, the cell suspension in PBS-Tx was centrifuged at 123× *g* for 8 min with no deceleration to generate a single-cell suspension. The single-cell suspension was diluted to OD_600nm_ = 0.1 in a total volume of 30 mL of PBS-Tx. Colony forming units/mL (CFUs) were determined by plating serial dilutions of the cell suspension in duplicate on 7H11 Middlebrook media. The remaining culture was incubated without shaking at 37 °C for 2 weeks. Aliquots from the starved cultures were assessed for DCTB by MPN and CFU analysis. In addition, the starved cells and cells resuscitated from the MPN assay (DCTB) were stained with metabolic probes (fluorescent D-amino acids) and analyzed using a fluorescent microscope for differences in cell length and staining patterns as described in the following sections.

### 2.6. Assessment of Starved Cells for Differentially Culturable Tubercle Bacteria (DCTB)

The quantity of DCTB recovered after starvation of the *M. tuberculosis* strains was assessed using the most probable number (MPN) assay with and without culture filtrate (CF) and colony forming units as previously described [1].

#### 2.6.1. Preparation of Culture Filtrate

The culture filtrate (CF) was prepared as described previously [1]. Briefly, 1 mL freezer stocks of the *M. tuberculosis* laboratory strain (H37RV) and the five strains from either the Beijing family or LAM family were grown in 8 mL of 7H9 Middlebrook media to an OD_600nm_ = 0.5. The pre-culture was added to 42 mL of 7H9 Middlebrook media and grown to an OD_600nm_ = 0.6–0.8. Experiments with strains from each family were performed on separate days to prevent cross-contamination. In addition, a 1 mL aliquot of each culture was spread onto 7H11 plates to ensure the cultures were free of fungal and other bacterial contamination. The cells were harvested by centrifugation at 3000× *g* for 10 min. The supernatant was filtered using a sterile syringe and a 0.2 μm filter. To ensure the supernatant was free of bacteria, a 500 μL aliquot was spread onto 7H11 Middlebrook media (Sigma) and checked for mycobacterial contamination after 3 weeks of incubation. In addition, a 1 mL aliquot was incubated at 37 °C for 4–6 weeks and monitored for turbidity indicative of growth. The CF was prepared on the day of the experiment and diluted in a 1:1 ratio with 7H9 Middlebrook media, before using it in subsequent MPN assays.

#### 2.6.2. The MPN Assay

The MPN assay determines an approximate number of bacteria in liquid media using a limiting serial dilution and a Poisson distribution calculation. The limit of detection in the MPN assay is dependent on the dilution factor, number of dilutions set up and number of replicates. All MPN assays conducted in this study were performed in 96-well microtiter plates (LTC Tech) as described by Saito et al. [23]. Briefly, 180 µL of 7H9 Middlebrook media or CF was added to all the wells except the first row. The respective starved bacterial samples were vortexed 3 times for 10 s each to de-clump the cells, and 100 µL added to each well in the first row of the plate. The samples were mixed using a multichannel pipette and a 10-fold serial dilution was performed across the plate, by transferring 20 µL from the first row into the second row. The cells were mixed, and the process was repeated to the end of the plate. From the last row, 20 µL of the cells were removed and discarded to maintain equal volumes across the entire plate. The MPN plates were sealed and incubated at 37 °C for 5 weeks. All experiments were conducted with three biological replicates, and six internal replicates, to increase the accuracy of the MPN calculator. The MPN assay was also scaled in a 24-well plate format by adding 200 µL of the starved culture to 1,8 mL of media or CF, to ensure the sample volume detecting DCTB would be sufficient for further downstream analysis. After 5 weeks of incubation at 37 °C, the MPN plates were scored for growth using an inverted mirror. Wells with positive growth were recorded and the total number of bacteria recovered from the MPN assay was estimated using the MPN-calculator software available at (https://www.wiwiss.fu-berlin.de/fachbereich/vwl/iso/ehemalige/professoren/wilrich/MPN_ver6.xls, accessed on 18 January 2019).

#### 2.6.3. Determination of Colony Forming Units (CFUs)

Fifteen µL aliquots of dilutions (10^−1^ to 10^−5^) from each of the 6 internal replicates of the starved cells in media in the MPN plate were removed and placed in a new 96-well plate in the same order, for colony forming unit (CFU) assessment. For each dilution, the replicate aliquots were pooled and 25 µL from the 90 µL total volume was spread in duplicate on Middlebrook 7H11 plates and incubated at 37 °C for 3–5 weeks, after which CFUs/mL were calculated.

### 2.7. Resuscitation of Starved H37Rv, Beijing and LAM Cells in MGIT Supplemented with CF

Culture filtrate (CF) prepared either from H37Rv, Beijing or LAM strains was added to Mycobacterial Growth Indicator Tubes (MGIT, Becton-Dickenson) to assess if CF decreased the time to detection of starved *M. tuberculosis* (H37Rv) cells. Briefly, the media in the MGIT tube was diluted in a 1:1 ratio by removing 3.5 mL of media from the MGIT tubes, except for the control tube, and replacing it with 3.5 mL of either the H37Rv CF or CF produced from the clinical strains. Before inoculating the modified MGIT tubes with 500 µL of the starved H37Rv cells, 800 μL of PANTA reconstituted in OADC was added to the MGIT tubes. The tubes were incubated in the BACTEC MGIT 960 instrument, and the time to positivity was recorded. All experiments were performed in triplicate.

### 2.8. Metabolic Labelling of Axenic, Starved and Resuscitated Cells

After starvation, 1 mL of the starved culture was stained with 500 µM (2 µL) of TADA (Sigma) for 24 h at 37 °C and compared to stained cultures grown in 7H9 media. Post the staining, cells were harvested by centrifugation (3000× *g* for 10 min) and washed twice with an equal volume of 1× PBS. After washing, the cells were fixed with an equal volume of 2.5% glutaraldehyde (Sigma) for 3 h. The fixed cells were harvested by centrifugation (3000× *g* for 10 min), and washed twice with 1× PBS, before resuspension in 300 µL of 1× PBS. Similarly, cells resuscitated in media and CF were also labelled with TADA for comparative analysis. All experiments were conducted with three biological replicates in triplicate.

### 2.9. Cytological Assessment of Strained Bacteria by Microscopy

Five µL of labelled and fixed cells were spotted onto slides, covered with coverslips, and the edges sealed with adhesive. Microscopy was performed with a Nikon Eclipse T12 and PE4000 LED light source equipped with a Plan Fluor 100× oil immersion 1.30–numerical aperture objective. Images were captured in the TL DIC channel and TADA channel (maximum λ ex/em = 550/594 nm). The NIS-Elements AR software (Nikon Inc., Version 5.02, (Build 1266), Nikon Instruments Inc., Tokyo, Japan) and Image J (Fiji) software (http://imagej.net/Fiji, accessed on 14 March 2024, Version 1.54f, Wayne Rasband and contributors, NIH, USA, http://imagej.org) was used to process images. All image acquisition and processing was executed under identical conditions for both the control and samples.

### 2.10. Fluorescent Staining Patterns

A minimum of one hundred cells per replicate were visualized for the incorporation of the metabolic probe into the peptidoglycan (PG) layer of the cell wall, using a Zeiss Elyra super-resolution fluorescence confocal microscope (Zeiss, Jena, Germany). The slides were prepared as described above and viewed under the confocal microscope equipped with a Plan-apochromat 100× 1.46 oil aperture. Two sets of images were taken, and the first image was captured using a Confocal PLAM STORM application and included both a brightfield channel and the TADA channel (maximum λ ex/em = 550/594 nm). The second image was captured using the structural illumination super-resolution application with the following technical parameters: processing set at manual, mode set at 3D, noise filter set at −3.0, SR frequency weighting set at 1.0 and baseline cut sectioning set at 100, 95 and 95. The captured super-resolution images were processed using the structural illumination software from the Zen black and Image J (Fiji) software (http://imagej.net/Fiji, accessed on 14 March 2024, Version 1.54f, Wayne Rasband and contributors, NIH, USA, http://imagej.org). Each cell was analyzed for the presence of a fluorescent signal. Fluorescent signals were associated with 6 different staining patterns, which we assigned as follows: uniform staining, polar staining, polar and septal staining, side wall staining, uneven staining and strong and weak pole staining.

### 2.11. Cell Length Measurements

A minimum of 100 cells per replicate were visualized for cell length. Cell length was measured and recorded using the Nikon software (NIS–Elements AR) and Image J (Fiji) software (http://imagej.net/Fiji, accessed on 14 March 2024, Version 1.54f, Wayne Rasband and contributors, NIH, USA, http://imagej.org). The cell length data were compared relative to cells grown in 7H9. The mean, median and interquartile ranges were determined for each data set using Graphpad Prism 6.

### 2.12. Data Analysis

PowerPoint, Excel and GraphPad prism 6 software were used to generate the graphs and figures. GraphPad Prism 6 software was used for statistical analysis to compare significant differences between data groups.

## 3. Results

### 3.1. Quantification of DCTB in Sputum of Participants with Drug-Sensitive TB

The participant disposition flow chart for the cohort used in this study is shown in Figure 1A and the baseline participant demographics are provided in Table 1.

This cohort comprised participants from two previous primary studies, one aimed at investigating the role of DCTB as a measure of TB treatment response (BMG study) and the other investigating the role of CF supplementation in the detection of TB in TB-HIV co-infected individuals (MGIT-Plus study) [5,30]. A total of 109 participants were analyzed in this study with an average age of 36 years, and 62% of the cohort was male (Table 1). The majority (81%) of the participants were HIV-positive with an average cycle threshold of 21.2. The average MGIT time to positivity (TTP) was 8 days and sputum from 78% of the participants was smear-positive by Auramine staining (Table 1). In both these studies, the presence of DCTB in sputum was assessed by the limiting dilution MPN and CFU assays supplemented with CF or media, to identify CF-dependent or CF-independent DCTB, respectively (Figure 1B). Residual frozen sputum samples from 160 individuals from the two studies were spread on 7H11 plates. After 4 weeks of incubation at 37 °C, DNA was extracted from the cells and subjected to spoligotyping to categorize the samples into the various *M. tuberculosis* strain types. Strain categories with less than 10 representatives, and samples showing mixed infections or lack of genotypic detection by spoligotyping, were excluded from the analysis (n = 36). Samples with contamination or no detectable bacteria were also excluded (n = 15).

Spoligotyping of the DNA (Appendix A) from the remaining 124 samples indicated the presence of five major family types in our cohort which, in order of dominance, included T family (36/109, 33%), Beijing family (27/109, 25%), S family (18/109, 16.5%), LAM family (18/109, 16.5%) and X family (10/109, 9%) (Figure 1A). In the 124 specimens chosen for analysis, the presence or absence of DCTB was assessed by proportional analysis of the MPN data, as determined in the two previously published studies [5,30]. This analysis indicated that all strain types had the ability to produce DCTB, with the Beijing family (85%) and X strains (70%) harboring a greater proportion of DCTB. The T strains (47%), S family (56%) and LAM family (50%) showed an almost equal proportion of specimens with and without DCTB. Chi-squared T-test analysis showed that T strains, S family, LAM family and X strains had a significantly lower propensity to produce DCTB compared to the Beijing family (Figure 1C). The data were further analyzed to assess the effect of CF on the resuscitation of DCTB in the various *M. tuberculosis* strains. The addition of CF significantly enhanced the detection of DCTB in all the strain categories except in the Beijing family, where both media and CF resuscitated an equivalent number of bacteria (Figure 1D). Collectively, these data suggest that the Beijing family is the most adapted to generate DCTB and correspondingly produces the largest quantity of CF-independent DCTB compared to other clinical strains.

### 3.2. Generation of DCTB Using an In Vitro Carbon Starvation Model

Given that the combined analysis of DCTB data showed differential propensity of *M. tuberculosis* strains to generate DCTB during TB pathogenesis in humans, we aimed to further interrogate this observation. For this, 5 independent strains each from the Beijing family and from the LAM family were selected from the cohort and the genotype of the 10 strains was re-confirmed by spoligotyping before further downstream analysis (Appendix A). In addition, the growth kinetics for each of the 5 strains from both the Beijing and the LAM family were compared to those of H37Rv. All 5 strains of the Beijing family showed similar growth kinetics to each other, which were also comparable to H37Rv (Appendix A). The 5 LAM family strains showed comparable growth to each other but had a slower growth rate compared to H37Rv; however, this difference was statistically insignificant (Appendix A).

After confirmation of the genotypes, the carbon starvation model was used to induce an in vitro DCTB state. Exponentially grown bacteria were starved in phosphate-buffered saline (PBS) supplemented with tyloxapol for 2 weeks before resuscitation in nutrient-rich media (CF-independent) and nutrient-rich media supplemented with CF (CF-dependent) (Figure 2A).

Aliquots of the starved cells were stained with TADA, a fluorescent metabolic stain that incorporates into the peptidoglycan (PG) layer of the cell wall of actively replicating cells only. As a result, this probe serves as a proxy for mycobacterial replication and/or cell wall turnover. Assessment of approximately 300 H37Rv stained cells under the different culture conditions showed a major reduction in the proportion of stained cells in the starved culture compared to cells grown in nutrient-rich (7H9) media, suggesting reduced metabolic activity (Figure 2B,C). A similar ratio of stained versus unstained cells was observed for Beijing and LAM strains assessed under the same culture conditions of starvation and resuscitation (Appendix A). Resuscitation of starved cells with and without CF for each of the strains restored metabolic activity, as the number of stained cells detected under both these conditions was comparable to that of the culture grown in nutrient-rich media (Figure 2C and Appendix A). Interestingly, H37Rv starved cells showed a drastic reduction in cell length (*p* ≤ 0.0001) compared to cells grown in 7H9 media (Figure 2B,D). Resuscitation of the starved cells not only restored the cell length, but the cells were longer, particularly when resuscitated with CF (*p*-value ≤ 0.0001), compared to cells grown in 7H9 media (Figure 2B,D). In contrast, comparison of cell length differences for the un-starved, starved and resuscitated cells for the clinical strains showed that the starved cells from the Beijing family and LAM family were significantly longer (*p* ≤ 0.0001), compared to the resuscitated cells which were marginally longer than the cells grown in nutrient-rich media (*p* ≤ 0.05) (Appendix A). The LAM strains showed no significant change in cell length for cells resuscitated with CF (Appendix A).

Following starvation, the DCTB population was enumerated by resuscitation with media and CF generated from H37Rv. Strains from both the Beijing family and LAM family showed the propensity to enter the differential culturable (DC) state under nutrient starvation conditions (Figure 3A). The H37Rv and the Beijing strains generated similar DCTB proportions. Grouped analysis indicated that significantly more DCTB were resuscitated in both the Beijing and LAM strains with CF compared to resuscitation with media. Furthermore, a two-way ANOVA test showed that individual Beijing strains had a greater propensity to produce DCTB compared to the LAM strains (*p*-value ≤ 0.01) (Figure 3A).

Assessment of the average DCTB resuscitation population from the five strains, for both the Beijing and LAM genotype families, demonstrated that the Beijing strains produced more DCTB than LAM or H37Rv (Figure 3B). The Beijing strains were also able to produce more CF-independent DCTB when compared to the LAM strains, suggesting that they are not solely reliant on CF for resuscitation. This finding reflects the DCTB data from sputum (Figure 1D), indicating that our laboratory model of DCTB was able to replicate phenotypes that are prevalent in clinical specimens.

### 3.3. The Effects of Cognate CFs Derived from Beijing and LAM Strains in the Resuscitation of DCTB

The effect of CF derived from the Beijing and LAM strains compared to CF generated from H37Rv to resuscitate DCTB (generated from starved H37Rv) was interrogated using the MGIT BACTEC 960 system. This was conducted to determine the ability of different CFs to resuscitate DCTB. Half the media from the commercial MGIT tubes was removed and replaced with either 7H9 media (control), H37Rv-CF, Beijing-CF or LAM-CF. Growth of inoculated starved H37Rv cells was monitored as MGIT time to positivity (TTP) (Figure 4A).

CF from the Beijing and LAM families significantly decreased the TTP for starved H37Rv compared to cultures supplemented with media or H37Rv-CF (Figure 4B). In each case, there was about a 10 h difference in TTP between the strains. Two strains out of the five for both the Beijing family (strains Beijing 1 and Beijing 2) and LAM family (strains LAM 1 and LAM 5) were the most effective at resuscitating DCTB (*p* ≤ 0.01 and *p* ≤ 0.0001). Since CF from both the Beijing and LAM strains had the capacity to resuscitate the growth of starved H37Rv cells, we next assessed the role of the cognate CF from the Beijing strain (Beijing 2) to resuscitate DCTB in the five distinct starved Beijing strains (Figure 4B). The Beijing-CF resuscitated a greater amount of DCTB in all five strains compared to H37Rv-CF and the media control. However, this difference was only significant in the Beijing 1 and Beijing 4 strains (*p* ≤ 0.001 and *p* ≤ 0.01, respectively, Figure 4C). Similarly, the resuscitation of DCTB in the five independent starved LAM strains when assessed with LAM-CF (from LAM-5), H37Rv-CF and media, showed a marginal increase in resuscitation with the LAM-CF, and again only two LAM strains (LAM 4 and LAM 5) showed significant resuscitation (*p* ≤ 0.01) of DCTB compared to the resuscitation with H37Rv-CF and media (Figure 4D).

### 3.4. Cytological Profiling of Axenic, Starved and Resuscitated Cells by Fluorescent Microscopy

Axenic and starved cells of H37Rv, Beijing and LAM, resuscitated with 7H9 media and H37Rv-CF werestained with the TADA probe, and analyzed for staining patterns using SR-SIM fluorescence microscopy. Six different cell staining patterns were observed under the different growth conditions. These included uniform staining, polar staining, polar and septal staining, side wall staining, uneven staining, and strong and weak pole staining (Figure 5A and Appendix A).

The proportion of the six staining patterns varied under the different culture conditions for each of the strains. For H37Rv, most cells (22–36%) showed uniform and polar staining (Figure 5B) under all culture conditions. Six percent of the cells grown under nutrient-rich conditions showed side wall staining, which increased to 12% and 16% for starved cells resuscitated with media and CF, respectively (Figure 5B). Polar and septal staining was reduced in starved cells resuscitated with CF compared to those resuscitated with media. The proportion of uneven staining and strong and weak pole staining was approximately equivalent for cells cultured under all three conditions (Figure 5B). In contrast, a majority of the Beijing starved cells resuscitated with CF showed an uneven staining pattern (62%), with uniform staining and polar staining reduced by 20–26% compared to cells resuscitated with media and cells grown in 7H9 media, respectively (Appendix A). Beijing cells grown under all three culture conditions showed a similar proportion of cells with strong and weak pole staining (Appendix A). Like H37Rv, the starved LAM cells resuscitated with media or CF showed similar proportions of uniform and polar staining, but showed greater uneven staining, 22% versus 12% for H37Rv (Appendix A). LAM cells grown in 7H9 media showed a greater proportion (30%) of side wall staining compared to the resuscitated cells (8–12%), with a similar proportion of cells with strong and weak pole staining under all three culture conditions (Appendix A).

During replication, mycobacterial cells extend their poles by incorporation of new cell wall material at the polar region. As a result, the PG synthetic machinery is localized to cell poles during elongation, which will result in the incorporation of the TADA probe in this area. Given the variations in staining patterns observed between the clinical strains, we next set out to determine whether new PG incorporation is affected by the presence of CF and if there were any strain-specific incorporation patterns during resuscitation. TADA staining distribution plots of exponentially grown and resuscitated bacteria of H37Rv, Beijing and LAM were generated to determine the general pattern of probe incorporation along the cell wall. Our data showed that H37Rv, when grown in 7H9 or under resuscitation culture conditions, mainly showed polar staining (Figure 5C). The Beijing strain, when grown in 7H9, showed a mixture of polar and uniform staining, with the latter pattern maintained during the resuscitation of starved cells (Figure 5D). The LAM cells showed a similar staining pattern as H37Rv, predominately with staining at the poles under all three culture conditions (Figure 5E). Although live cell microscopy was not carried out, staining of the cell wall can be used as a surrogate for tracking cell growth. Further analysis of the staining patterns confirmed that when all three strains were cultured in 7H9 media, growth was occurring via polar extension, with some uniform staining (Figure 5F). Starved cells resuscitated with media (7H9) showed growth from the cell pole for LAM, whilst H37Rv and Beijing incorporated the probe along the entire cell wall surface (Figure 5G). Resuscitation of the starved cells with CF showed polar growth for all three strains (Figure 5H).

## 4. Discussion

Beijing strains are associated with hyper-virulence and increased transmission in the human population; however, little is known about their ability to produce DCTB [26,32,33,34]. We therefore carried out a meta-analysis of the DCTB data from two previous clinical studies to investigate the propensity of different clinical strains to generate DCTB. Sputum containing the Beijing strain had the highest quantity of DCTB, suggesting that Beijing strains are more robust at adopting the differentially culturable state when compared to other clinical strains. The reason for this may be linked to the increased expression of the DosR regulon genes under stress conditions, compared to non-Beijing strains, to better protect and poise the mycobacteria for survival [28,35]. Although evidence points to Beijing strains having greater virulence and transmission potential compared to other clinical strains, this phenomenon appears to be inter-strain dependent as there is a high amount of heterogeneity amongst Beijing strains [33,36,37,38,39,40].

Recently, variant analysis, transcriptional studies and genome-wide transposon sequencing were used to unravel the biological basis of the distinctive phenotypes of Beijing strains. These analyses identified functional genetic changes across multiple stress and host response pathways, suggesting that these adaptive changes may be the reason for the distinct clinical characteristics and epidemiological success of this *M. tuberculosis* family [41].

Traditionally, bacterial culturability has been defined as growth on solid media; however, in 2010, Mukamolova and colleagues demonstrated that 80 to 99.9% of viable *M. tuberculosis* in sputa were undetectable by standard culture assays [2]. The reasons for this deficiency in culturability may be twofold: (i) the organism may encounter more damaging conditions during pathogenesis and expectoration that prevent favorable growth, and (ii) metabolic changes during infection may provide a selective advantage allowing protection from the host immune system, resulting in DCTB [1,2,42]. Several studies have provided evidence that previously undetected populations of bacteria on solid media can be detected in liquid media with enhanced growth in the presence of CF [1,3,4,5,23,43,44]. Starved mycobacteria in the DCTB state have been shown to have increased tolerance to antibiotics as the transcription of various genes is selectively affected to induce the differentially culturable state [23]. As described by Saito et al., we used an increased rifampicin (RIF) concentration (100× MIC), aiming to circumvent impaired uptake of RIF by starved cells or phenotypic RIF resistance resulting from stress-induced protein mistranslations [23]. Under these conditions, we were unable to induce increased DCTB but instead observed killing of the mycobacteria [45]. Treatment of starved cells with low concentrations of RIF (25× MIC) confirmed sensitivity to RIF, observations which were in concordance with Betts et al. who showed a 3.5 log drop in the CFU count of starved cells treated with 10× the MIC of RIF [15].

We randomly selected five Beijing and five LAM strains together with the laboratory strain (H37Rv) to robustly generate metabolically distinct non-replicating mycobacteria using the carbon starvation model. Subsequently, we established the ability of these starved bacteria, in a DCTB state, to resuscitate in the presence or absence of CF. Beijing strains produced more DCTB compared to LAM under carbon starvation conditions. LAM strains appeared to require CF for resuscitation whilst Beijing DCTB showed almost equivalent resuscitation with or without CF supplementation. This observation suggested that nutrient deprivation stress induces distinctive metabolic shifts in different mycobacterial strains requiring specific resuscitation conditions [1,42]. To measure cell wall turnover in DCTB, a metabolic probe specific for PG synthesis was used. Microscopic analysis of stained carbon-starved bacteria confirmed reduced metabolic activity, indicative of bacteria entering a non-replicative state. Starved H37Rv cells were significantly shorter compared to those from a culture grown to exponential phase. Morphological changes in mycobacteria during non-replicating persistence have been reported. During stationary phase acidification (pH 8.5 → 4.7) of culture media due to cellular metabolism, resulted in the formation of phase-dark ovoid cells with thickened cell walls, low metabolic activity and elevated resistance to heat and antibiotics [46]. These ovoid cells lost the ability to form colonies on solid medium and regained culturability only with CF from active *M. tuberculosis* cells, or an externally added recombinant form of Rpfs [46]. Resuscitation of the starved cells with media restored the cell length to that of cells grown in axenic culture and they were marginally longer when resuscitated with CF. This morphological change may be due to differences in cell extension rates as bacteria emerge from the DCTB state. Here, the presence of CF may allow for faster cell elongation, a hypothesis that would require time-lapse analysis for confirmation.

Starved Beijing cells also showed a significant increase in the number of resuscitated bacteria with CF from the cognate strain compared to CF from the laboratory strain, H37Rv. In contrast, starved LAM strains did not show enhanced resuscitation with its cognate CF, and the amount of DCTB detected was equivalent to that with CF from H37Rv. These data suggest that the Beijing strains possibly require strain-specific molecules to exit the DC state. Moreover, the reduced requirement by Beijing strains for CF during resuscitation confirms that these bacteria can spontaneously resuscitate without the need for exogenous growth factors. This, combined with their increased propensity to form DCTB, may partly explain the epidemiological success of these strains.

The gold standard for TB diagnosis is a culture of sputum in the MGIT system, which detects the growth of *M. tuberculosis* as a measure of oxygen consumption over time. Since a MGIT positive culture test can take between 2 and 6 weeks, attempts to shorten the detection time are desirable for improved diagnostic turnaround. In a recent study, the addition of CF derived from *M. tuberculosis* H37Rv to MGIT media decreased the time to positivity (TTP) in HIV-infected patients with a low bacterial load [30]. Thus, we compared the utility of H37Rv-CF, Beijing-CF and LAM-CF to enhance resuscitation of H37Rv DCTB. Both Beijing-CF and LAM-CF showed improved resuscitation of DCTB compared to the H37Rv-CF, as the TTP was reduced on average by approximately 10 h. These observations have significant implications in shortening the time to TB diagnosis and improving treatment outcomes.

Metabolic staining of starved cells for both the laboratory and clinical strains showed reduced uptake of the cell wall probe, indicative of a non-replicative state. Resuscitation of starved cells with media or supplementation with CF restored the metabolic activity comparable to logarithmically growing cultures for H37Rv and LAM. Analysis of the incorporation of new cell wall material during the resuscitation of starved cells using a cell wall probe identified six different cell wall staining patterns in varying proportions for the three strain genotypes. As expected, polar and uniform incorporation of the peptidoglycan, indicative of growth, were the predominant patterns observed for all three strains under normal and resuscitation growth conditions. The differences in staining patterns observed for the Beijing and LAM strains merit further study as they point to possible metabolic differences in clinical strains when exposed to stressful conditions.

Collectively, our data demonstrate that both Beijing family and LAM family strains can enter the DCTB state under carbon starvation conditions, with Beijing family strains having a greater propensity for this, suggesting that certain strains of *M. tuberculosis* are more adept to enter the non-replicating state under stress conditions. Future studies should focus on the mechanistic basis that underpins the difference in the propensity of the Beijing and LAM strains to adopt the differentially culturable state.

## Figures and Tables

**Figure 1 pathogens-13-00318-f001:**
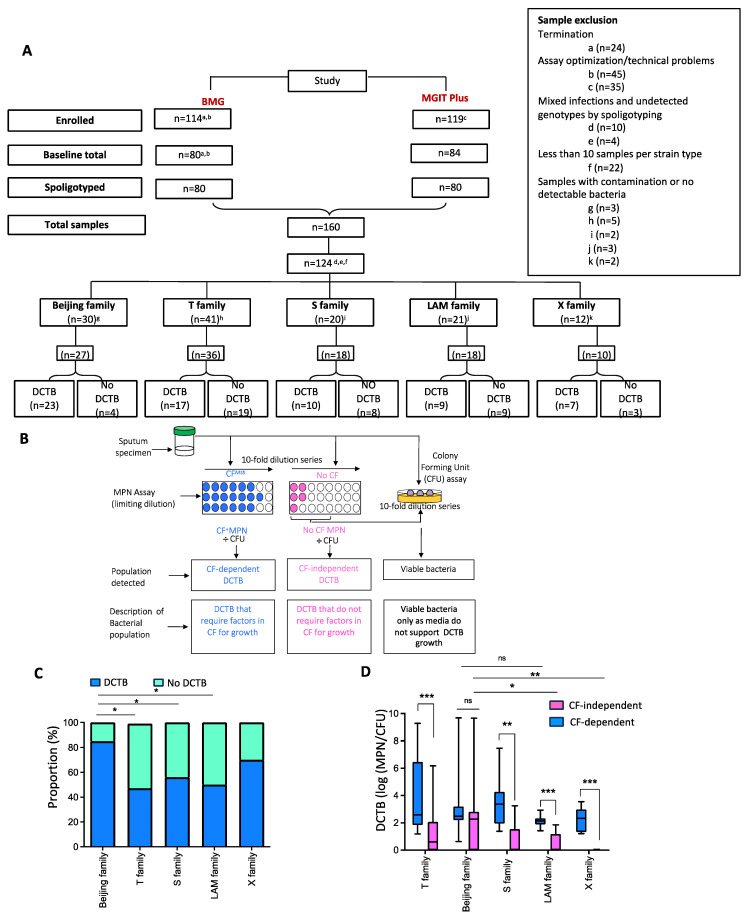
Participant disposition flow chart and assessment of DCTB in baseline sputum samples. (**A**) Participant disposition flow chart for individuals recruited in the BMG and MGIT Plus studies for the DCTB data analysis. Samples excluded for various reasons are indicted with the letters a-k. The Beijing family belongs to lineage 2 (East-Asian) whilst lineage 4 (Euro-American) includes the T, S, LAM and X families. (**B**) Schematic diagram adapted from Gordhan et al. [4] shows the methodology used for DCTB assessment in sputum specimens in real time obtained at baseline. Sputum samples were decontaminated and assessed by CFU and the Most Probable Number (MPN) limiting dilution assays containing culture filtrate (CF) to detect DCTB. To control for the effect of CF in growth stimulation, fresh Middlebrook 7H9 media was used (No CF MPN) which allowed for quantification of conventionally culturable bacteria. DCTB ratios were obtained by dividing the MPN values (with or without CF) by CFU counts. (**C**) Proportion of DCTB detected in the different *M. tuberculosis* genotype families. DCTB had to be present in only one of the conditions that is with or without CF, within each family. Genotype families with less than 10 samples were excluded from the analysis. The Beijing family showed the greatest amount of DCTB (*p*-value ≤ 0.05) using a chi-squared t-test, compared to the other families. (**D**) Assessment of the effect of CF on the resuscitation of DCTB in the various *M. tuberculosis* families. The amount of DCTB was calculated as log (MPN/CFU) for both the CF-independent and CF-dependent DCTB for each strain type. Using the Mann-Whitney test (GraphPad), the Beijing family showed significant resuscitation with CF (*p*-value < 0.05) and more DCTB (*p*-value = 0.0518) were detected within this family compared to the other genotype families. * = *p*-value < 0.05, ** = *p*-value < 0.01, *** = *p*-value < 0.001 and ns = not significant.

**Figure 2 pathogens-13-00318-f002:**
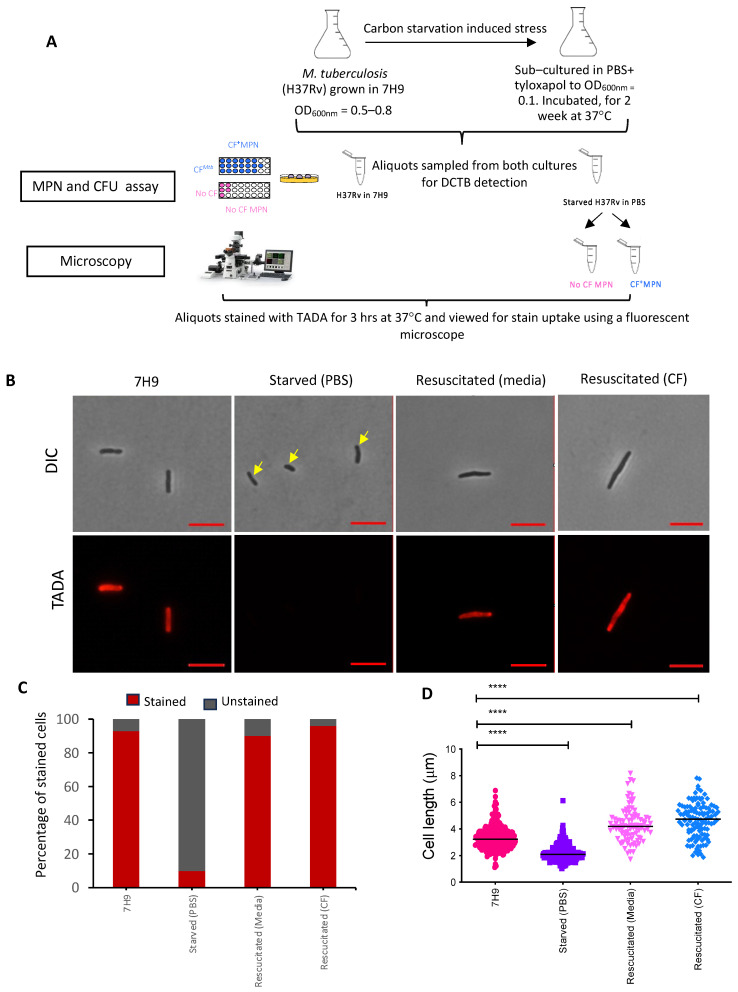
Generation of DCTB in H37Rv using an in vitro carbon starvation model. (**A**) Schematic diagram showing the conditions for carbon starvation of H37Rv and downstream assessment for the detection of DCTB. The DCTB population in H37RV grown in 7H9 media and starved in PBS was detected by the MPN and CFU assays, followed by the assessment of the resuscitated cells (with media and CF), starved and axenic cells by microscopy. (**B**) Microscopy of stained starved cells before and after resuscitation with and without CF compared to an axenic culture of H37Rv grown in 7H9 media. (**C**) Assessment of culturability of cells under the different culture conditions. Up to one hundred TADA-stained cells from each of the culture conditions were assessed for the proportion of stained and unstained cells under a fluorescent microscope. (**D**) Comparison of cell length differences for the unstarved, starved and resuscitated cells. The scale bar represents 2 µM. A one-way ANOVA test showed that the starved H37Rv cells were significantly shorter (**** = *p*-value ≤ 0.0001) compared to the resuscitated cells which were significantly longer than the cells grown in 7H9 (**** = *p*-value ≤ 0.0001).

**Figure 3 pathogens-13-00318-f003:**
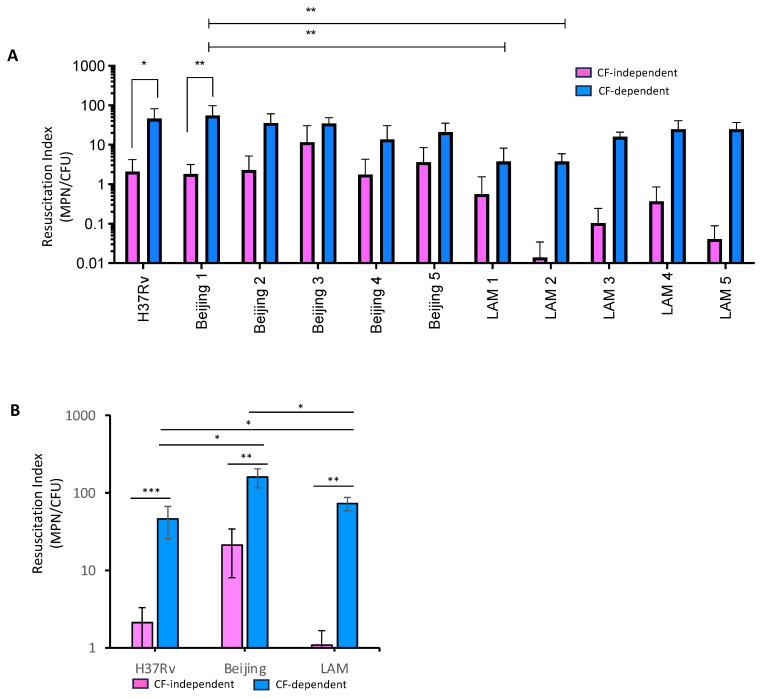
Assessment of the propensity of the clinical Beijing and LAM isolates to produce DCTB. (**A**) Each of the 5 Beijing and LAM strains and the H37RV control laboratory strain were starved in PBS for two weeks, after which DCTB from each culture were resuscitated with media and CF from H37Rv. The resuscitation index was calculated using the MPN/CFU ratio. The experiment was performed in triplicate. *T*-test analysis showed that both Beijing and LAM strains resuscitated significantly more CF-dependent DCTB compared to resuscitation with media (*p*-value ≤ 0.01). (**B**) Average DCTB resuscitation index for the 5 Beijing and 5 LAM strains. * = *p*-value < 0.05, ** = *p*-value < 0.01 and *** = *p*-value < 0.001 All three genotype families demonstrate greater resuscitation of the starved strains with CF compared to media (*p*-value ≤ 0.001–≤0.01) with no difference in resuscitation between the Beijing and LAM families (*p*-value ≤ 0.01).

**Figure 4 pathogens-13-00318-f004:**
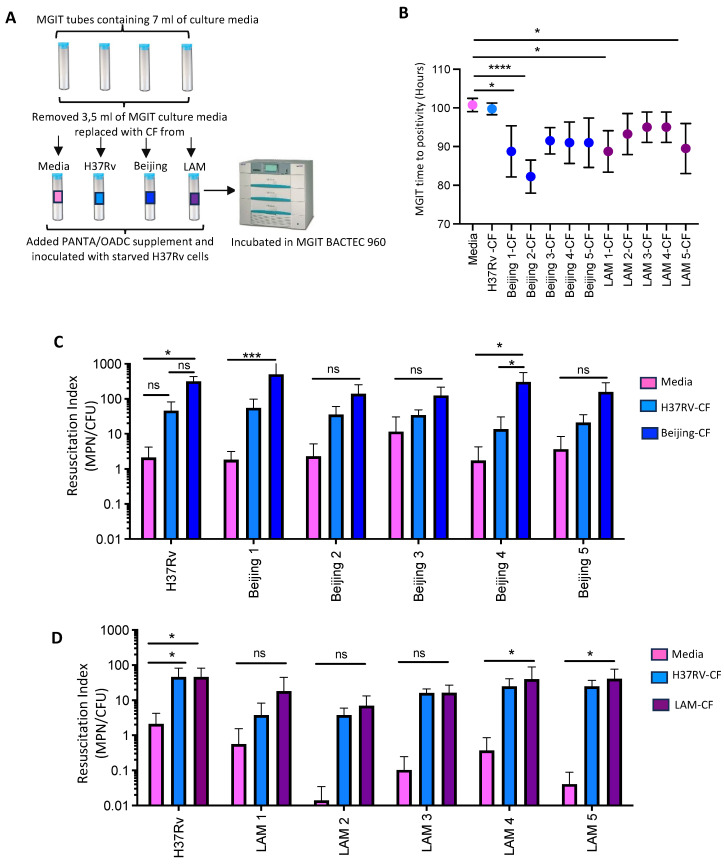
The effects of CFs derived from clinical isolates in the resuscitation of DCTB. (**A**) A schematic representation of the procedure followed to supplement MGIT tubes with CF from the Beijing and LAM clinical strains. (**B**) MGIT time to positivity (TTP) of starved H37Rv supplemented with Beijing-CF and LAM-CF compared to CF from H37RV and media. A one-way ANOVA test showed that CF from both Beijing and LAM strains significantly decreased TTP compared to media and H37Rv-CF (* = *p*-value ≤ 0.05 and **** = *p*-value ≤ 0.0001). (**C**) Resuscitation Index of the five starved Beijing strains compared to starved H37Rv, resuscitated in media, H37Rv-CF and Beijing- CF (from Beijing 2). A two-way ANOVA statistical test showed that Beijing-CF allowed greater detection of DCTB in starved Beijing 1 and Beijing 4 cells when compared to the media control (* = *p*-value ≤ 0.05, *** = *p*-value ≤ 0.001 and ns = not significant). (**D**) Resuscitation Index of five independent starved LAM strains compared to starved H37Rv, resuscitated in media, H37Rv-CF and LAM-CF (from LAM 5). A two-way ANOVA statistical test showed no significant resuscitation of DCTB with LAM-CF compared to starved cultures resuscitated with media and H37Rv-CF. Two of the LAM strains showed significant resuscitation with LAM CF compared to media (* = *p*-value ≤ 0.05 and ns = not significant). Resuscitation index was plotted as the MPN/CFU ratio. The experiments for each strain were performed in triplicate.

**Figure 5 pathogens-13-00318-f005:**
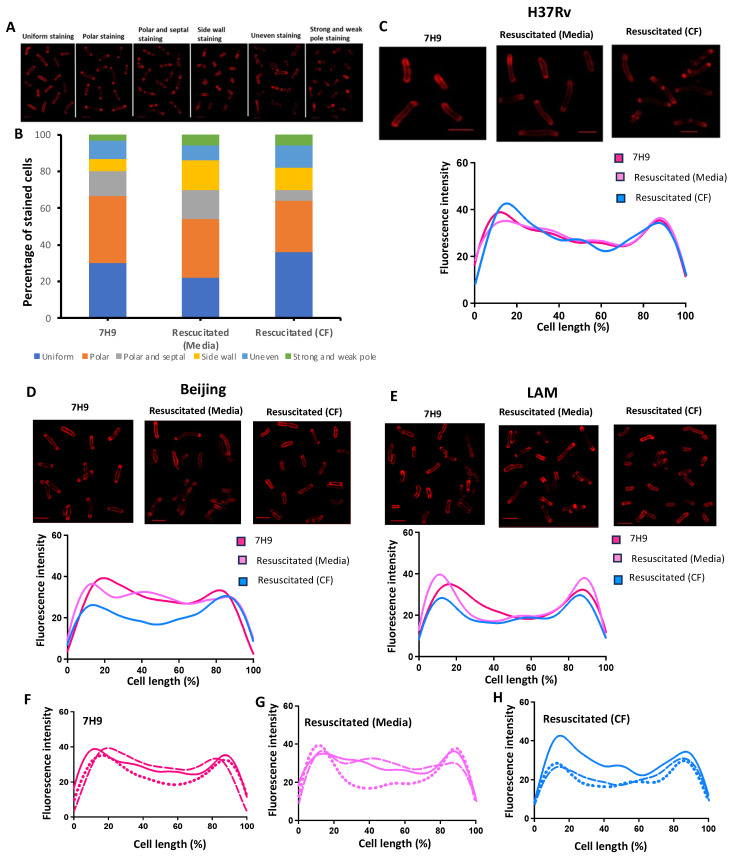
Cytological profiling of axenic, starved and resuscitated cells by fluorescent microscopy. Starved cells were resuscitated in 7H9 media and CF, and stained with the TADA probe for 24 h at 37 °C. The experiment was performed in triplicate. One hundred cells from each condition were analyzed for staining patterns using a SR-SIM fluorescent microscope. (**A**) Microcopy showing the presence of six different staining patterns for H37RV resuscitated cells. Scale bars represent 2 µM. (**B**) The proportion of cells resuscitated with media and CF, showing the distribution of the six identified cell staining patterns. (**C**) Comparison of TADA staining distribution plots of exponentially grown axenic culture of H37RV and resuscitated bacteria. The pattern of probe incorporation along the poles and cell wall was assessed for 30 cells under each condition. Similar staining patterns were assessed for 30 axenically grown and resuscitated Beijing cells (**D**) and LAM cells (**E**). The staining pattern for the 30 cells for the three strains were further analyzed under nutrient- rich culture conditions (**F**), resuscitated with media (**G**) and resuscitated with CF (**H**). In each case, the solid line represents H37Rv, the dashed line represents the Beijing strain, and the dotted line is the LAM strain.

**Table 1 pathogens-13-00318-t001:** Demographics of study participants.

	Study		Combined Studies
	BMG Study (n = 61)	MGIT PLUS Study (n = 48)	BMG Study and MGIT PLUS Study (n = 109)
**Demographics**			
Sex: Female (%) Male (%)	34 66	42 58	38 62
Age, yrs, Median (IQR)	35 (26.5–45.0)	37 (30–44)	36 (28.5–44.0)
BMI, Median (IQR)	19.07 (17.8–22.1)	21.5 (18.4–23.5)	20.17 (18.0–22.8)
**HIV Status, n (%)** Negative Positive	34 66	0 100	19 81
**GeneXpert cycle threshold (IQR**)	18.7 (14.3–24.4) *	22.7 (18.7–26.6) ^#^	21.2 (16.3–25.2) ^§^
**Conventional TB Diagnosis, n (%)** Smear grade negative Smear grade positive	23 77	21 79	22 78
**MGIT Time to Positivity, d (IQR)**	6 (4.0–11.5)	11.0 (7.3–15.0)	8.0 (5.0–14.5)

* GeneXpert test results negative for 2 participants; ^#^ GeneXpert test results negative for 4 participants; ^§^ GeneXpert results negative for 6 participants.

## Data Availability

The raw data supporting the conclusions of this article will be made available by the authors on request.

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
