# Peer review of "Clinical Strains of *Mycobacterium tuberculosis* Representing Different Genotype Families Exhibit Distinct Propensities to Adopt the Differentially Culturable State"

_pathogens, 2024, doi:10.3390/pathogens13040318_

Round 1

Reviewer 1 Report

Comments and Suggestions for Authors

Congratulate the authors for the work done.

The only thing I would highlight is that in lines 93 and 94, which correspond to the introduction, the authors present conclusions derived from the results obtained that should go in the conclusions section.

Author Response

Comments from reviewer 1

We thank the reviewer for taking the time to review this manuscript and appreciating the work. Please find the response to the comment below and the corresponding revisions/corrections are highlighted/in track changes in the re-submitted files.

Point-by-point response to Comments and Suggestions for Authors

Comment 1: The only thing I would highlight is that in lines 93 and 94, which correspond to the introduction, the authors present conclusions derived from the results obtained that should go in the conclusions section.

Response 1: Thank you for pointing this out. We agree with this comment and have therefore removed this statement from the introduction and added it to the discussion section.

Reviewer 2 Report

Comments and Suggestions for Authors

The work carried out and presented in the article is of very good quality. It represents a significant interest for the scientific community working on tuberculosis. The objective and the methodology are clearly explained. The results are presented with clarity and relevance. The text is well-written and coherent.

One small remark: I find it surprising to have conducted spoligotyping on the strains while currently, everyone uses Whole Genome Sequencing (WGS) to analyze/genotype the strains. Genome analysis could help to go deeper into the study. However, the study is already quite comprehensive for this publication. Nevertheless, it is an idea for further research on the topic.

Author Response

Comments from Reviewer 2

We thank the reviewer for taking the time to review this manuscript and appreciating the work. Please find the response to the comment below 

Point-by-point response to Comments and Suggestions for Authors

Comment 1: One small remark: I find it surprising to have conducted spoligotyping on the strains while currently, everyone uses Whole Genome Sequencing (WGS) to analyze/genotype the strains. Genome analysis could help to go deeper into the study. However, the study is already quite comprehensive for this publication. Nevertheless, it is an idea for further research on the topic.

Response 1: Thank you for pointing this out. We agree that the field has moved to whole genome sequencing (WGS) for genome analysis. We are set up to carry out spoligotyping but not WGS. Since this was a student project, spoligotyping formed part of the training. Limited funding also prohibited WGS. Moreover, for the purposes of this study we  only required classification of the stains and not an in depth analysis of the strain genotypes. As suggested our current studies have incorporated WGS for in depth analysis of strain specific genotypes. . 

Reviewer 3 Report

Comments and Suggestions for Authors

In this study, the authors have evaluated clinical isolates of M. tuberculosis strains belonging to different genotype families, and  demonstrated  genotype family-related variations in the ability of the strains  to enter into and resuscitate from a Differentially Culturable state. The study findings can have implications on more accurate Mtb diagnosis and treatment. The study methods are clearly presented and appropriate for the research question. 

While the manuscript presents reasonable data to support the study conclusion, there are some overarching issues that should be addressed prior to the publication:

1. Title: The title must reflect that the distinct propensities are attributed to  Mtb strains representing different genotype families.

2. The entire Manuscript: the Authors should consider a more accurate use of the terminology when referring to Mtb strains that belong to different genotypes. Such, the simplistic description of the study objects  as “Beijing and LAM clinical strains” (L22) from the very beginning is somewhat unsuitable. The Authors discriminate clinical Mtb isolates according to their spoligotypes, therefore the established nomenclature  should be used (family, lineage, or sublineage).  For example,  the “LAM strains” have to be described as strains that belong to a LAM family genotype at the beginning of the manuscript prior to application of a more concise strain reference style throughout the text. 

3. The entire Manuscript: the Authors should  reconsider an interchangeable use of “strains” and “isolates”. The Authors establish their findings based on evaluation of strains that are genetically different by definition. The isolates might be not genetically different, and therefore the more accurate use of terminology  is encouraged.

4. The Materials and Methods Section is missing brand information for multiple materials. Please provide.

5. L105: Please re-evaluate the terminological application of the phrase “in silico” to define the type of the work performed in the study. Some of the data have been generated in wet lab (not in silico) experiments conducted in previous studies, and are being presented in the context of the currently reported study. The “in silico assessment ” creates a false impression, and also does not correctly describe the work done. I assume that analysis and interpretation of previous data have not been reported, and therefore can be rightfully used and reported in the Manuscript. 

6. L110-112. Please proofread the statement “mycobacteria isolated from these participants, using the Most Probable Number (MPN) assay”. The MPN assay is normally used for bacterial number determination,  but not for isolation of  mycobacteria from human sputum. 

7. L112-113: Spoligotypes of clinical strains,  that were included  in the present study (see Fig 1),   must be provided. This requirement could be addressed  in a form of a supplementary table  with the list of isolates with associated spoligotype binary codes, spoligotype lineages, and spoligotype  families, and DCTB categories. 

8. Figure 1A, B and C: Please reconsider the style used for identification of strain genotypes (currently they are T-strains, S-family, or simply as ‘Beijing”) (see the Comment 2).

9. L117: Please specify what Mtb strain/type was used for CF preparation and use.

10. L171: Please clarify the “… distinct Beijing and 5 distinct LAM isolates ”. What makes these isolates distinct? Also address the application of more specific definition for LAM and Beijing family genotypes. 

11. L210-211: How the supernatant cultures were evaluated for contamination? Please provide methods used.

12. L301: “In silico Quantification” – please reconsider the “in silico” as per the previous related comment. 

13. L336: Please reconsider the wording in “… combined in silico analysis”. As per my above comments, the use of “in silico” is unsuitable, as data were derived from previous unpublished wet lab experiments. 

14. Fig 2. Please verify the graphs. Specifically, the Y axes  legends suggest log MPN/CFU values, but the data are also  plotted in logarithmic scales. Overall, the value range 0.01 to 1000 seems odd, and the resuscitation indexes are presented incorrectly.

15. Fig 4C, D.   Please verify the graphs. Specifically, the Y axes  legends suggest log MPN/CFU values, but the data are also  plotted in logarithmic scales. Overall, the value range 0.01 to 1000 seems odd, and the resuscitation indexes are presented incorrectly.

16. L448-L450. Please critically evaluate the statement “The Beijing strain when grown in 7H9 showed polar staining, a pattern which was maintained during resuscitation of starved cells (Figure 5D).” The Figure 5D contradicts this statement and demonstrates a different resuscitation pattern in media conditions. This Authors’ statement is also contradicts the further statement in L455-457. 

17. L493-500: The purpose of the provision of this information is unclear and confusing, as the information seems to be irrelevant to the study results discussion and the manuscript per se.

18. Figure S1. Please reconsider the data presentation in the tables and the strain abbreviations. I suggest in incorporate additional columns in the tables to provide: a) strain abbreviations as per the Manuscript (Beijing 1, LAM 1 etc., see Fig 3 and 4); b) replace the current “Strain” in the table column headings with “ Spoligotype Lineage”, c) add a column with the corresponding spoligotype family. Also please coordinate the abbreviation of the strains used in the study: the growth rate plots provide data on strains defined as B1 and L1, etc, while the Manuscript describes strains identified as Beijing 1, LAM 1, etc. (see Fig 3 and 4).

19. L551-555: Please evaluate the statement that starts with “As expected, polar and uniform …” , as this conclusion partially contradicts an earlier provided statement in the L448-458.

20. L517-520: Please provide literature references for the statement “The ovoid cells … form of Rpfs.”

Other comments:

21. L66: “Betts and co-workers developed” – I suggest to relocate the Ref#15 here from the end of the sentence.

22. L79-80: The introduction of abbreviations MDR and XDR is not necessary, as these abbreviations are not used further in the text.

23. L86: Correct “… that this strains is well adapted …” to “ … that these strains are well adapted …”

24. L93: Correct “both Beijing and LAM have” to “both Beijing and LAM family strains have”

25. L94: Correct “with Beijing having a greater propensity” to “with Beijing family strains having a greater propensity”

26. L106: Please clarify the “BMG and MGIT- plus”. Are these studies designated as  BMG and MGIT- plus?

27. Fig 1B: Please proofread the “spot sputum specimen”. The “spot” is not commonly used terminology.

28. L127: Please correct “The unique spacer DNA sequence…” to “The unique spacer DNA sequences…” as 43 spacer oligonucleotide types are evaluated in the spoligotyping procedure.

29. L129: Correct “was amplified” to “were amplified”.

30. L131-132: Please proofread the statement (should it be “each PCR tube”? ), also it looks overly descriptive.

31. L131-132: Proofread the statement. Was the membrane hybridised at this step? 

32. L158: “(Beijing and LAM)”. Please consider  “Beijing and LAM genotype families”

33. L174: Please elaborate “ the various mycobacterial strains” . Are these Mtb strains? What are the genotypes and the total number?  

34. L185: “Beijing and LAM” – please elaborate as per the Comment #3. 

35. L187: Please provide the volume of the phosphate buffer saline.

36. L 190-191: “…total volume of 30  ml and the colony forming…”. Please clarify what is the 30 ml refers to, and insert a comma.

37. L208: Amend “were harvested at 3000 ×g for 10 minutes” to “were harvested by centrifugation at 3000 ×g for 10 minutes”.

38. L291: “A minimum of 100 cells” – the Fig 2C legend states “up to 300 cells” . Please harmonise the description of the cell range in the text and the Fig 2C legend.

39. Figure1B Legend: Amend “Schematic” to  “Schematic diagram”.

40. Figure1B Legend: Please proofread “DCTB counts were obtained by dividing…”. Was the ratio, but not counts, obtained?

41. L306: “(BMG)”- please clarify. Is it the Study BMG?

42. L308: “(MGIT-Plus)” – please clarify. Is it the Study MGIT-Plus?

43. L318-334: Please apply an established nomenclature  of genotype families.

44. Fig 2 Legend: The “Schematic showing…”  can be corrected to “Schematic diagram showing…”

45. L375: Although the “DC state” is understandable, the  “DC” abbreviation has to be formally introduced.

46. L390-391: Please proofread and reconsider the sentence, and it is confusing due to potentially missed words or commas.

47. Fig 2B Legend: Please proofread and correct the statement “All three starved strains show greater resuscitation …” . In fact,  all three strain genotypes demonstrate the greater resuscitation of starved strains (there were more than 3 strains in the study).

48. L401: Consider changing “…both Beijing (isolates 1 and 2), and LAM (isolates 1 and 5) were…” to “…both Beijing (isolates 1 and 2), and LAM (isolates 1 and 5) family genotypes were…”.

49. Fig 5 legend: The Legend is unfinished.

50. L467: Please correct “…the DOS regulon…” to “…the DosR regulon…”

51. L472-473: Please proofread the sentence for punctuation.

52. L475: Please proofread the sentence for punctuation.

53. L482: Please proofread the sentence for punctuation.

54. L510: “peptidoglycan (PG)” – the PG abbreviation has been introduced earlier in the Manuscript and therefore is unnecessary here.

55. L517: Please proofread the punctuation.  

56. L550: Correct “…the three strains” to “…the three strain genotypes”, as more than three strains were used in the study.

Comments on the Quality of English Language

Minor suggestions are provided in the comments for Authors.

Reviewer 4 Report

Comments and Suggestions for Authors

This study uses in vitro carbon starvation model to determine the propensity of Beijing and LAM clinical strains to generate DCTB. The quantitation of DCTB would be helpful to find out the source of recurrence, rate of relapse/refection states and also to discover biomarker to assess treatment response, drug resistance. Authors strictly followed the appropriate methods and demonstrated the results without bias. The authors should address the following comments to improvise the quality of the manuscript.

1.      Authors mentioned the data was retrospectively from previous two studies. However, I would ask authors to add a table with demographics, clinical characteristics, and laboratory screening results in each strain family.

2.      Figure 1B has been adapted from reference 4. Indeed, the figure is almost identical. Although it is necessary to demonstrate the MPN Assay, it should be indicated that this is a figure published already in a previous article, it is up to the journal to decide whether a permission is necessary to reuse a figure from an open access article.

3.      What host immune response/factors drives bacteria into phenotypically distinct, drug-tolerant states DCTB population? It was proposed that DCTB population is dependent on the CD4 counts and other innate cells. But I do know there has been much discussion in the literature on how best to assess to understand this biological effect.

4.      It has been demonstrated that M. tb strains presents DCTB in sputum under various stress conditions that mimics in vivo conditions. I could have really appreciated if the authors could have approach in mechanistic evidence for the formation of DCTB. This could have help in develop therapeutic tools to abrogate the DCTB formation.

5.      Similar studies have conducted in same/different settings by using various clinical strains. Some of them are cited in this manuscript as well. The current manuscript makes a useful addition to that literature but does not represent a significant scientific advance on it.

Author Response

Comments from reviewer 4:

We thank the reviewer for taking the time to review this manuscript and for the suggestions to improve the manuscript. Please find the detailed responses below and the corresponding revisions/corrections highlighted/in track changes in the re-submitted files.

Comment 1: Authors mentioned the data was retrospectively from previous two studies. However, I would ask authors to add a table with demographics, clinical characteristics, and laboratory screening results in each strain family.

Response 1: A table describing the demographics for the participants (Table 1) from the two studies has been added with the associated text (L254-262)

Comment 2: Figure 1B has been adapted from reference 4. Indeed, the figure is almost identical. Although it is necessary to demonstrate the MPN Assay, it should be indicated that this is a figure published already in a previous article, it is up to the journal to decide whether a permission is necessary to reuse a figure from an open access article.

Response 2: Thank you for raising this. We have redrawn the figure and have referenced the previous article from our group that has a similar figure.

Comment 3: What host immune response/factors drives bacteria into phenotypically distinct, drug-tolerant states DCTB population? It was proposed that DCTB population is dependent on the CD4 counts and other innate cells. But I do know there has been much discussion in the literature on how best to assess to understand this biological effect.

Response 3: To address the reviewers comment, there has been a recent study by Saito et al.(referenced below) that demonstrated that M. tuberculosis cells enter the differentially culturable (DC) state when they experience sublethal levels of oxidative stress responsible for damaging the cellular DNA, proteins, and lipids. As a result the cell replication process is delayed, to allow for DNA and protein repair. In addition, they showed that Mycobacterium bovis and its derivative, BCG, was unable to enter the DC state under similar conditions suggesting that this phenomena may be unique to M. tuberculosis with implications for tuberculosis latency, detection, relapse, treatment monitoring, and drug resistance. Given that we did not measure immune function in this study, we opted not to include any mechanistic information. We have attempted in vitro DCTB models using some stresses, but our results were inconsistent. Hence, we feel it is not appropriate to delve into these matters in the manuscript.

Kohta Saito, Saurabh Mishra , Thulasi Warrier , Nico Cicchetti, Jianjie Mi , Elaina Weber, Xiuju Jiang, Julia Roberts, Alexandre Gouzy, Ellen Kaplan, Christopher D Brown, Ben Gold, Carl Nathan (2022). Oxidative damage and delayed replication allow viable Mycobacterium tuberculosis to go undetected. Sci Transl Med. 24;13(621).doi:10.1126/scitranslmed.abg2612

Comment 4: It has been demonstrated that M. tb strains presents DCTB in sputum under various stress conditions that mimics in vivo conditions. I could have really appreciated if the authors could have approach in mechanistic evidence for the formation of DCTB. This could have help in develop therapeutic tools to abrogate the DCTB formation.

Response 4: We agree with the reviewer that understanding the mechanism(s) for the formation of DCTB is important for the development of therapeutic tools to prevent the formation or eradication of DCTB. However, this aspect was not the focus of this study. Since our group has been involved for many years in studying DCTB phenomena, we are looking into this aspect of research in our future studies. 

Comment 5: Similar studies have conducted in same/different settings by using various clinical strains. Some of them are cited in this manuscript as well. The current manuscript makes a useful addition to that literature but does not represent a significant scientific advance on it.

Response 5: We agree with the reviewer that several studies from our group and others have investigated DCTB in both drug susceptible and drug resistant cohorts. These studies have highlighted the presence of DCTB at baseline, during treatment and at the end of treatment suggesting, that the current treatment regimens are ineffective against eradicating all DCTB. However, our current study to the best of our knowledge is the first where an attempt has been made to associate the different M. tuberculosis genotype families and their inherent propensities to generate DCTB. This information could contribute towards understanding the reasons behind TB treatment failure, transmission and disease relapse, when related to particular M. tuberculosis strains.

Round 2

Reviewer 3 Report

Comments and Suggestions for Authors

Thank you for addressing comments and suggestions provided in my first review. 

Comments on the Quality of English Language

Please proofread the the revised manuscript (specifically, the the added text) for typos. Examples: Lines 28, 51, 100, 116, etc. 

Reviewer 4 Report

Comments and Suggestions for Authors

There are few typos and grammatical errors in the revised manuscript. I hope this will be corrected in proofreading stage. 

Overall, the authors responded to my comments and revised the manuscript well. I think this manuscript will be acceptable for publication upon fixing the very minor corrections.